# Feasibility, acceptability and equity of a mobile intervention for Upscaling Participatory Action and Videos for Agriculture and Nutrition (m-UPAVAN) in rural Odisha, India

Emily Fivian[1]*, Manoj Parida[2], Helen Harris-Fry[1], Satyanarayan Mohanty[2], Shibanath Padhan[3], Ronali Pradhan[4], Pranay Das[2], Gladys Odhiambo[1], Audrey Prost[5], Terry Roopnaraine[6], Satyaranjan Behera[2], Philip James[7], Naba Kishor Mishra[3], Suchitra Rath[8], Nirmala Nair[8], Shibanand Rath[8], Peggy Koniz-Booher[9], Heather Danton[9], Elizabeth Allen[10], Suneetha Kadiyala[1]

1 Department of Population Health, London School of Hygiene & Tropical Medicine, London, United Kingdom, 2 DCOR Consulting Pvt. Ltd., Bhubaneswar, India, 3 Voluntary Association for Rural Reconstruction and Appropriate Technology (VARRAT), Kendrapara, India, 4 Digital Green, New Delhi, India, 5 Institute for Global Health, University College London, London, United Kingdom, 6 Independent Consultant, Valencia, Spain, 7 Emergency Nutrition Network (ENN), Oxford, United Kingdom, 8 Ekjut, Chakradharpur, India, 9 JSI Research & Training Institute, Inc. Arlington, Virginia, United States of America, 10 Department of Medical Statistics, London School of Hygiene & Tropical Medicine, London, United Kingdom

* emily.fivian@lshtm.ac.uk

**Data Availability Statement:** The de-identified participant quantitative data analyzed during the

## Abstract

Addressing undernutrition requires strategies that remove barriers to health for all. We adapted an intervention from the 'UPAVAN' trial to a mobile intervention (m-UPAVAN) during the COVID-19 pandemic in rural Odisha, India. In UPAVAN, women's groups viewed and discussed participatory videos on nutrition-specific and nutrition-sensitive agricultural (NSA) topics. In m-UPAVAN, weekly videos and audios on the same topics were disseminated via WhatsApp and an interactive voice response system. We assessed feasibility, acceptability, and equity of m-UPAVAN using a convergent parallel mixed-methods design. m-UPAVAN ran from Mar-Sept 2021 in 133 UPAVAN villages. In Feb-Mar 2021, we invited 1000 mothers of children aged 0–23 months to participate in a sociodemographic phone survey. Of those, we randomly sampled 200 mothers each month for five months for phone surveys to monitor progress against targets. Feasibility targets were met if >70% received videos/audios and >50% watched/listened at least once. Acceptability targets were met if >75% of those watching/listening liked the videos/audios and <20% opted out of the intervention. We investigated mothers' experiences of the intervention, including preferences for m-UPAVAN versus UPAVAN, using in-person, semi-structured interviews (*n* = 38). Of the 810 mothers we reached, 666 provided monitoring data at least once. Among these mothers, feasibility and acceptability targets were achieved. m-UPAVAN engaged whole families, which facilitated family-level discussions around promoted practices. Women valued the ability to access m-UPAVAN content on demand. This advantage did not apply to many mothers with limited phone access. Mothers highlighted that the UPAVAN interventions' in-

current study are available for ethically approved research from the LSHTM Data Compass, an open access institutional research data repository: https://doi.org/10.17037/DATA.00003635. The qualitative data analyzed during the current study contains unique participant experiences that may risk revealing participants' identities, even after the removal of common identifiers. Given the nature of the discussions, this may risk jeopardizing participant safety. Data are available from the corresponding author, or the data management team at the London School of Hygiene and Tropical Medicine via email (researchdatamanagement@lshtm.ac.uk) for specific lines of enquiry that do not jeopardize the participants' anonymity and safety, and subject to ethical approval.

**Funding:** This work was co-funded by the Bill & Melinda Gates Foundation and the Foreign Commonwealth Development Office of the UK Government (award number OPP1136656) to EF, MP, HHF, SM SP, RP, PD, GO, AP, SB, PJ, NKM, SuR, NN, ShR, PKB, HD, EA, and SK. Additionally, funding of author HH-F was provided by a Sir Henry Wellcome Fellowship (grant 210894/Z/18/Z). The funders had no role in study design, data collection and analysis, decision to publish, or preparation of the manuscript.

**Competing interests:** I have read the journal's policy and the authors of this manuscript have the following competing interests. AP declares receiving funding from the Children's Investment Fund Foundation, The UK Medical Research Council, and The UK Economic & Social Research Council; payment for grant reviews from the Research Council of Norway; payment for guest lecturing at the London School of Hygiene and Tropical Medicine; and being the Chair of Climate Change Working Group as part of the Children in All Policies (CAP2030) Initiative. These competing interests will not alter adherence to PLOS Global Public Health policies on sharing data and materials. No other authors have competing interests to declare.

person participatory approaches and longer videos were more conducive to learning and inclusive, and that mobile approaches provide important complementary. We conclude that mobile NSA interventions are feasible and acceptable, can engage families, and reinforce learning. However, in-person participatory approaches remain essential for improving equity of NSA interventions. Investments are needed in developing and testing hybrid NSA interventions.

## Introduction

Maternal and child undernutrition remains a leading cause of death and disability in low- and middle-income countries [1]. In India, decades of government programmes have aimed to address this burden, yet millions still suffer. The UN Global Strategy for Women's and Children's Health expresses the need to implement and scale up existing cost-effective, evidence-based interventions and for innovative strategies that eliminate barriers to these interventions [2]. Innovative strategies include mobile technologies that could increase access and efficiency of health care delivery, especially in resource-constrained, remote settings [3,4]. However, a prerequisite for mobile health interventions to reach and benefit people is that participants have mobile phone access. This means that mHealth risks widening inequalities in access to health services rather than improving them. While mobile phone access is almost universal among men in India, around half of women do not have access and those that do tend to access lower quality phones, such as feature phones rather than smartphones [5]. Women's phone access is also constrained by gendered social barriers that affect their phone usage, yet women are often key targets for health interventions due to specific antenatal and maternity healthcare needs and the role they commonly play in ensuring children's healthcare access [6]. Additionally, the feasibility of mHealth in resource-constrained areas may also be challenged by poor network connectivity, digital illiteracy, misinformation and cybercrimes, phone numbers changing frequently and the economic inability of poor households to recharge phones [7,8].

Alongside potential challenges, mobile approaches may present opportunities to overcome barriers that arise from face-to-face approaches. For example, mobile interventions that are accessible from home and on-demand can overcome time-related barriers (e.g., travel times to health centres or women's groups, and time away from domestic responsibilities and livelihood activities), restrictions to women's physical mobility [9], and reluctance to access services when heavily pregnant or with children [10]. Further, mobile approaches may be more convenient for other family members to engage with, which may enhance the adoption of promoted practices compared with interventions solely targeting women [6]. Despite mobile interventions being increasingly used to improve healthcare delivery in India, there is a lack of well-designed user acceptability and efficacy studies [11]. Additionally, little is known about whether mHealth intervention coverage and engagement would be sufficient and the trade-offs between mobile and in-person approaches. Therefore, a better understanding of the feasibility, acceptability, inclusivity and equity implications of adapting effective health interventions to mHealth delivery is needed.

One type of effective intervention that may be conducive to mHealth adaptation are the nutrition-sensitive agriculture (NSA) interventions tested in the four-arm cluster-randomised controlled trial, UPAVAN [12]. The UPAVAN interventions used combinations of women's group facilitated viewings and discussions on locally made participatory NSA and nutrition-specific videos, and a cycle of participatory learning and action meetings. The trial was

conducted in 148 clusters (village and surrounding hamlets) in four blocks of Keonjhar district, Odisha, India. Keonjhar is a heavily forested landlocked district where 45% of the population belongs to historically disadvantaged groups, Scheduled Tribes, who endure high rates of poverty and undernutrition [13,14]. The UPAVAN study arms included:

AGRI: Fortnightly women's group meetings using NSA participatory videos;

AGRI-NUT: Fortnightly women's group meetings using NSA videos once per month and nutrition-specific participatory videos once per month;

AGRI-NUT+PLA: Women's group meetings using participatory NSA videos once per month and nutrition-specific participatory learning and action meetings or videos once per month;

Control: Standard government services only.

The interventions ran for 32 months between 2016–19. The impact evaluation showed a higher proportion of children achieving minimum dietary diversity in AGRI-NUT and AGRI-NUT+PLA compared to the control group, and a higher proportion of mothers achieving minimum dietary diversity in AGRI and AGRI-NUT+PLA compared to the control group [12]. Since UPAVAN's completion in Oct 2019, the COVID-19 pandemic caused disruptions to livelihoods and many essential face-to-face nutrition and health services [15]. These developments provided an urgent impetus to learn how communities can continue to benefit from agriculture and nutrition interventions despite the challenges posed by the pandemic.

Given this and community demand for the continuation of UPAVAN, we designed a behaviour change intervention retaining the core UPAVAN model with locally made NSA and nutrition-specific videos but adapted for mobile delivery (m-UPAVAN). The design of the m-UPAVAN intervention was informed by considering several of these potential opportunities and barriers, in addition to key learnings from the UPAVAN process evaluation [10]. Our study objectives were:

1. To assess the feasibility and acceptability of the m-UPAVAN intervention among mothers with a child 0–23 months.

2. To assess the extent to which the m-UPAVAN intervention was equitable by assessing feasibility and acceptability across specific subgroups.

3. To understand mothers' receptivity of m-UPAVAN contrasted against the face-to-face group-based UPAVAN approach.

## Materials and methods

### Ethical statement

Our study was approved by the London School of Hygiene and Tropical Medicine Ethics Board (#22800) and Sigma Institutional Review Board, New Delhi (#10039 and #10036). All implementation and research activities were conducted in concordance with local government guidance related to COVID-19.

### Sampling participants

m-UPAVAN included a subset of 99 (133 villages) of the 148 UPAVAN clusters in all four blocks of the UPAVAN trial (Ghatgaon, Harichandanpur, Patna, and Keonjhar Sadar) in Keonjhar district, Odisha, India. All 37 UPAVAN control clusters were included and intervention clusters with a higher proportion of Scheduled Tribe households were prioritised.

Local intervention facilitators visited households in all 133 villages between Dec 2020 and Jan 2021 to obtain data on phone ownership and phone numbers and identify eligible

participants for feasibility and acceptability assessment (women with a child under 2 years and intra-household phone access). Facilitators sought consent to share phone numbers with the implementation and research teams. All women in m-UPAVAN villages were invited to participate in m-UPAVAN using their own phone, a neighbour's, or a facilitator's phone when COVID-19 restrictions allowed.

**Quantitative phone survey sampling.** First, from the list of eligible participants, women and their spouses were randomly selected and invited to participate in a phone-based sociodemographic baseline survey between Feb 16 and March 7, 2021. Our target sample size was 1,000 mothers and their spouses. Next, following rollout of the m-UPAVAN intervention, a random sample of mothers was drawn each month from those who participated in the baseline survey between May and September 2021. These mothers were invited to participate in phone-based monitoring surveys designed to assess intervention feasibility and acceptability. Our target sample size was 200 mothers each month. Sampling from the baseline enabled us to link cases and avoid repetition of demographic questions. Following local government guidance on COVID-19, all but the first monthly surveys were conducted via phone. Therefore, we excluded the first month's survey (conducted in April) from analyses to avoid bias arising from different data collection methods. For phone surveys, data collectors sought informed verbal consent from participants, which was recorded and documented by DCOR–this was approved by the Institutional Review Boards. All participant selection was stratified by cluster-level distance to the nearest town ($<$10 km or $\geq$10 km) and proportion of Scheduled Caste and Scheduled Tribe households ($<$30%, 30% to 69%, $\geq$70%).

The number of intervention clusters and sample sizes for quantitative surveys was determined by budget and feasibility and deemed sufficient by our research team to describe feasibility and acceptability.

**Qualitative interview sampling.** After the intervention, between Oct 11 and 20, 2021, we recruited 38 women with a child under 2 years for qualitative interviews using purposive sampling to reflect the caste and tribe diversity of the population and varying phone access to enable us to explore a variety of responses to the intervention. Thirty-three women were sampled from those participating in monitoring surveys and, therefore, had intrahousehold phone access, of which we ensured at least 10 of these women had exposure to the previous UPAVAN interventions. Five women without intrahousehold phone access were also selected. Our qualitative sample characteristics are shown in **S1 Table.** Qualitative interviews were administered in-person, and data collectors sought participants' written (signature or thumbprint) informed consent.

**The m-UPAVAN intervention.** Digital Green, an international NGO, produced the intervention content. Voluntary Association for Rural Reconstruction and Appropriate Technology (VARRAT), an Odisha-based NGO, was responsible for implementing the intervention in m-UPAVAN villages. Ekjut and JSI provided technical assistance. UCL advised on the intervention content and study design. DCOR led the data collection and data coding. LSHTM led the entire study. These same organisations were partners in the UPAVAN trial.

The m-UPAVAN intervention was centred around weekly WhatsApp videos and audio messages on agriculture and nutrition topics, coordinated by intervention facilitators. The key components of the intervention are as follows:

1. Recruitment and training of facilitators

Intervention facilitators, most of whom were facilitators in the UPAVAN trial, were recruited at the village level. They were trained through Digital Green's Virtual Training Institute, a compilation of training resources to orient all staff on the essential nutrition, agriculture, and health topics addressed in the intervention. Facilitators were responsible for

community sensitisation, encouraging participation, and managing WhatsApp groups for video dissemination.

2. Dissemination of re-purposed UPAVAN videos through WhatsApp

The UPAVAN interventions built upon a participatory video-based approach to agriculture extension designed by Digital Green but made 'nutrition-sensitive' (i.e., promoting practices that address the underlying causes of undernutrition and incorporating specific nutrition goals) and integrating nutrition-specific video topics [16]. In m-UPAVAN, Digital Green reproduced videos covering NSA and nutrition-specific topics based on those used in the original UPAVAN trial but shortened for sharing and viewing via WhatsApp groups. Topics were sequenced based on season and community preference, as informed by experience from the UPAVAN trial [12]. On average, videos were 4 minutes long, with critical messages ('knowledge recall points') written in the local language (Odiya) at key points in videos, and 49 seconds of COVID-19 guidelines were included at the end of each video. Facilitators formed village-level WhatsApp groups for disseminating videos and 'WhatsApp viewers groups' where women who owned a smartphone were grouped with women who did not own a smartphone to watch videos together. Facilitators shared one video per week on Wednesdays over 6 months.

3. Interactive Voice Response (IVR)

For those owning a feature phone only, the same content as in the weekly videos was scripted as audio and disseminated every Sunday through a toll-free IVR system. Each audio was available to listen to until the next was shared the following week. Users could respond to the audio or ask questions by submitting a voice recording.

4. Promotional activities and interactive learning

Facilitators shared short video advertisements through WhatsApp before each video dissemination, which highlighted the constraints that each topic addressed. Village-wise posters advertised the toll-free IVR system (**S1 Appendix**), and text messages were sent each Sunday to remind registered users to listen to the audio messages. Facilitators encouraged participants to watch or listen to m-UPAVAN messages and to watch with their families via WhatsApp group chat or in-person, depending on COVID-19 restrictions. Facilitators helped community members with queries and encouraged them to adopt new practices. After each video dissemination, a quiz with multiple-choice questions took place in WhatsApp groups. Participants who selected the correct answers received an appreciation message, whilst facilitators encouraged those who got the wrong answer to re-watch the video. Facilitators also encouraged women to share photos of families watching or listening together and of them implementing the recommended practices.

Facilitators contacted all mothers with children under two years of age and a random selection of households with a phone at least once per month to ascertain if they were watching or listening, receive feedback on the content for future dissemination, and update phone numbers.

**Quantitative outcomes.** Our four key quantitative outcomes were: 1) proportion of women receiving at least one video or audio in the last month (coverage); 2) proportion of women watching or listening to at least one video or audio in the last month (uptake); 3) proportion of women that watched or listened who reported liking the messages (acceptability), and 4) proportion of women opting-out of from receiving m-UPAVAN messages, as indicated by the implementation team (attrition).

A priori, we determined that feasibility would be confirmed with a coverage rate of >70% and uptake rate of >50%, and acceptability confirmed with an acceptability rate of >75% and attrition rate of <20%. Our research and implementation team arrived at a consensus for targets by considering: 1) exposure to the original UPAVAN interventions, which ranged from 50–58% [12] and reflects our m-UPAVAN intervention uptake target (proportion of women watching or listening to at least one video or audio in the last month) of >50%; 2) potential logistical constraints, and 3) what the implementation team considered realistic, achievable and would warrant further investment, based on their >30 years of experience in implementing development interventions in our study district. As suggested by Avery and colleagues [17], indicators falling below our prespecified targets would be reviewed by the implementation team to assess whether further investment in the intervention was still warranted but with caution and or/modifications, or whether the indicators suggest intractable issues with the intervention that are irremediable.

Other outcomes included intensity of uptake, proportion of spouses or senior family members also watching or listening, and proportion of women discussing content with an adult family member.

Equity was assessed by analysing differences in all outcomes across well-known proxies of socioeconomic position in Odisha: whether participants reported belonging to *Scheduled Tribes* (ST) or other caste groups (non-ST), whether women had or had not completed primary school, and also by mother's personal phone ownership, and whether there was a smartphone in the household.

**Qualitative interviews.**   Interviews were conducted to understand women's experiences of the intervention, factors that affected its feasibility and acceptability, and the potential for realising change in NSA and nutrition-specific practices. Participants from the UPAVAN intervention areas were asked whether they participated in the UPAVAN interventions, and if so, questions on preferences for m-UPAVAN *vs* UPAVAN were additionally administered. All in-depth interviews were semi-structured, with open-ended questions and relevant prompts. Interview guides were piloted and refined, as necessary, and were conducted in Odiya at participants' homes by five experienced qualitative researchers from DCOR (see **S2 Appendix** for interview guides). Interview recordings were translated and transcribed verbatim into the English language by DCOR before analysis.

## Analysis

**Quantitative data analysis.**   Quantitative analyses were conducted in Stata/SE 17.0. Participants' characteristics are presented using descriptive statistics (frequency (%) and means (standard deviation)). We calculated coverage, uptake, and acceptability to assess achievements against the pre-specified feasibility and acceptability targets across all monitoring surveys using mixed-effects logistic or linear regression to obtain percentages or means (95% CI). To assess equity, we calculated adjusted percentage point (pp) differences across characteristics and phone access using mixed-effect logistic regression models followed by Wald tests for differences in post-estimation average adjusted marginal predictions, and mean differences in outcomes using mixed-effect linear regression models. Models included random effects to account for repeated measures within individuals and within clusters.

**Qualitative data analysis.**   NVivo 10 was used to manage and analyse qualitative data. We analysed interview data using deductive thematic analysis and used an adapted version of Normalised Process Theory (NPT) as a theoretical lens to assist with analysis and to organise and code the data. NPT "identifies, characterises and explains key mechanisms that promote and inhibit the implementation, embedding and integration of a new health technique,

technologies and other complex intervention" [18]. NPT is primarily used to evaluate intervention implementation or feasibility focusing on the agency of those involved in the implementation. Therefore, we adapted the four main constructs (coherence, cognitive participation, collective action, and reflexive monitoring) to reflect participants' experiences [19]. We divided interview transcripts among three researchers (MP, PD, GO). A fourth researcher (EF) double-coded one in three transcripts; EF reviewed inconsistencies, which were discussed with the team and resolved.

**Integration of quantitative and qualitative data.** To understand the feasibility and acceptability of the intervention beyond what we would from the qualitative and quantitative components separately, we used a convergent parallel mixed-method design [20]. We assessed concordance between the quantitative and qualitative findings and used the qualitative to explain the quantitative findings. We report findings using a narrative-weaving approach.

**Reflexivity and positionality statement.** This research was primarily undertaken by academic researchers based in the UK and researchers based in and from India. These researchers all have first-hand-contextual experience of the study setting and academic knowledge that likely preconditioned beliefs related to participant realities, which may have influenced interactions with participants during qualitative interviews and interpretations of the data. By acknowledging the diverse positionalities and perspectives within our research team and in relation to our study participants throughout the research processes, we strived to ensure a respectful representation of the lived experiences of the study participants.

**Inclusivity in global research.** Additional information regarding the ethical, cultural, and scientific considerations specific to inclusivity in global research is included in the (**S1 Checklist**).

## Results

In the 133 villages enrolled in m-UPAVAN, facilitators formed one WhatsApp group per village. All women in the villages owning a smartphone ($n = 3,099$) formed video viewing groups with 7–8 women without a smartphone ($n = 18,679$). 5,667 households were registered to receive IVR audio services. Intervention activities were delivered as planned. An overview of the intervention's content with video links is shown in **S2 Table**.

**Fig 1** shows the study flow diagram, and **Table 1** shows the characteristics of the quantitative survey participants. We identified 3,012 women with a child aged under 2 years in intervention villages. Of these, 27% ($n = 808$) were not eligible for quantitative surveys due to not having intra-household phone access. Of those that did have intra-household phone access, 1043 mothers provided data at baseline (cohort for monitoring surveys); 61% ($n = 632$) belonged to households owning a smartphone, and 67% ($n = 694$) of mothers owned their own phone (whether feature or smartphone).

Between May and September 2021, 810 mothers out of 1043 who provided data at baseline were randomly drawn to participate in monitoring surveys at least once. Of these, 666 mothers provided data (981 total responses). Differences existed between those who responded to monthly monitoring surveys and those who did not ($n = 114$). Compared with non-respondents, a larger proportion of respondents had a smartphone in the household (percentage point (pp) difference: 19.8) and owned ≥2.5 acres of land (pp difference 14.1).

Pooled descriptive results for outcomes across the five monthly monitoring phone surveys are provided in **Table 2.** No one opted out of receiving the intervention (attrition).

### Feasibility

The proportion reporting receiving at least one video or audio message in the last month was 72.8% (95% CI 69.1, 76.6), meeting our coverage target of 70%. Two qualitative themes

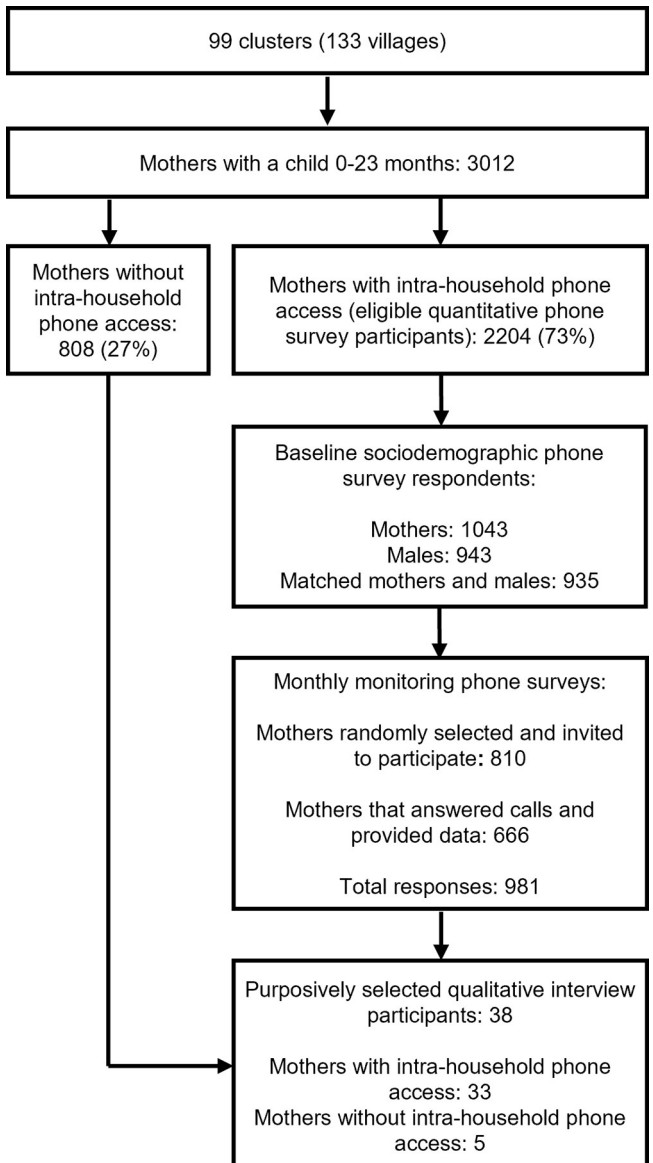

**Fig 1. Study flow diagram.** All quantitative surveys were administered by phone; qualitative interviews were administered in-person. Mothers randomly selected and invited to participate in monthly monitoring phone surveys indicate the total number of mothers that were randomly drawn and contacted at least once across five months of sampling.

emerged to explain the high coverage: trust in facilitators and perceived value of the intervention. Respondents were willing to participate in the intervention due to their trust and respect for the facilitators who informed them about the intervention, which assured participants that the intervention was worthwhile and relevant to them.

> *Respondent [R]: [. . .] As this program is good and Madam [facilitator] recommended to join I did the same. I didn't have to ask anyone's permission.*
>
> *R: I asked them [facilitators] where will the videos be sent and why. Then they answered that videos related to pregnant mothers and related to agriculture would be sent [. . .]. I got*

**Table 1. Characteristics of baseline sociodemographic and monthly monitoring phone survey participants.**

| | Baseline sociodemographic sample N = 1043 | | Monthly monitoring phone surveys (May-Sept 21) | | | |
| --- | --- | --- | --- | --- | --- | --- |
| | | | Respondents N = 666 | | Non-respondents N = 144 | |
| Characteristic | N | Mean (SD) or *n* (%) | N | Mean (SD) or *n* (%) | N | Mean (SD) or *n* (%) |
| **Social group, n (%)** | 1043 | | 666 | | 144 | |
| Scheduled Tribe | | 520 (49.9) | | 325 (48.8) | | 75 (52.1) |
| Other caste groups | | 523 (50.1) | | 341 (51.2) | | 69 (47.9) |
| **Mother's age in years, mean (sd)** | 1043 | 24.3 (4.4) | 666 | 24.6 (4.6) | 144 | 23.9 (4.1) |
| **Mother completed less than primary school, n (%)** | 1043 | 317 (30.4) | 666 | 207 (31.1) | 144 | 47 (32.6) |
| **Number of phones in the household, mean (sd)** | 1043 | 2.0 (1.1) | 666 | 2.0 (1.16) | 144 | 1.8 (1.1) |
| **Smartphone in the household, n (%)** | 1043 | 632 (60.6) | 666 | 432 (64.9) | 144 | 65 (45.1) |
| **Mother owns a phone, n (%)** | 1043 | 694 (66.5) | 666 | 464 (69.7) | 144 | 91 (63.2) |
| Size of landholdings, n (%) [a] | 934 | | 602 | | 119 | |
| <2.5 acres | | 686 (73.5) | | 422 (70.1) | | 101 (84.9) |
| ≥2.5 acres | | 248 (26.5) | | 176 (29.2) | | 18 (15.1) |
| Low asset score, n (%) [a] | 935 | 292 (31.2) | 602 | 180 (29.9) | 120 | 50 (41.7) |
| **Times participated in monitoring surveys, mean (sd)** | - | - | 666 | 1.8 (0.8) | - | - |

Note: Low asset score is those falling below the population mean household asset score derived from a sum of 14 assets asked in the male's baseline survey. Less than primary school are mothers with <8 years of schooling. [a] Data missing due to no matched males baseline sociodemographic survey.

*attracted because we are farmers so if they will send videos related to agriculture it will be beneficial to us as we can learn new things.*

**Table 3** shows that coverage was higher among those with smartphones in the household than those without (adjusted pp difference (95% CI): 9.4, 3.2, 15.6) and among women owning their own phones than women not owning their own (adjusted pp difference: 9.5, 2.7, 16.3). Qualitative findings revealed that whilst some spouses were keen to participate in the intervention and receive the m-UPAVAN messages on their phones, others needed more convincing.

*Interviewer [I]: What was your husband's reaction regarding you watching the videos on his phone? R: He is very happy to receive the videos on his phone. He says that after I finish all my house chores, we both should watch it together.*

*R: First they [spouse and mother-in-law] were very sceptical about it [m-UPAVAN] but later when I explained to them that it is important, then they accepted.*

The proportion reporting watching or listening at least once in the last month was 70.9% (95% CI 66.7, 75.1), satisfying our prespecified target for uptake of 50%. Women generally displayed agency in accessing and using phones. When women had to ask permission, they were rarely denied. Furthermore, spouses and other household members tended to support women's participation and facilitated their access to the m-UPAVAN content.

**Table 2. Acceptability, feasibility and other outcomes from all monthly monitoring phone surveys, May-Sept 21.**

| Indicator | | Crude | | Adjusted for survey design |
|---|---|---|---|---|
| | N | Mean (SD) or *n* (%) | | Mean or % (95% CI) |
| *Feasibility* | | | | |
| Coverage: Received videos or audios at least once, n (%) | 981 | 721 (73.5) | | 72.8 (69.1, 76.6) |
| Uptake: watched or listened at least once, n (%) | 981 | 710 (72.4) | | 70.9 (66.7, 75.1) |
| *Acceptability* | | | | |
| Liked the video a lot or a bit [a], n (%) | 585 | 585 (100.0) | | 100.0 (100.0, 100.0) |
| Liked the audio a lot or a bit [b], n (%) | 281 | 281 (100.0) | | 100.0 (100.0, 100.0) |
| *Other outcomes* | | | | |
| *Intensity of uptake* | | | | |
| Total videos or audios watched or listened to, mean (sd) | 981 | 1.64 (1.25) | | 1.61 (1.50, 1.72) |
| Total videos watched, mean (sd) | 981 | 1.38 (1.31) | | 1.36 (1.25, 1.47) |
| Total audios listened to, mean (sd) | 981 | 0.54 (0.94) | | 0.52 (0.45, 0.59) |
| *Engaging family members* [c] | | | | |
| Spouse or senior family member also watched or listened, n (%) | 710 | 429 (60.4) | | 60.8 (56.3, 65.3) |
| Discussed video or audio content with adult family member, n (%) | 710 | 366 (51.6) | | 51.5 (47.6, 55.4) |

Note: The recall period for all outcomes for each survey round is the previous 4 weeks. Adjusted for survey design use models which include random effects for repeat-measures within individuals (level 1) and within clusters (level 2). [a] Denominator includes participants that reported watching at least one video. [b] Denominator includes participants reporting listening to at least one audio. [c] Denominator includes participants reporting watching or listening to at least one video or audio.

*I: You have to ask for the mobile phone from your husband? R: No, no. He himself says that a message has come and I have to listen. I don't have to ask. I: Is he present at home when the message comes? R: Wherever he is, he comes home in the evening and says that message has come"*

The proportion of respondents watching or listening at least once in the last month was 7.2 (95% CI 0.28, 14.1) pp higher among women who owned a phone than women who did not (**Table 3**) but did not vary across other characteristics explored. One woman described her exchange with her spouse, explaining how his taking of the only phone for his work limited her access to the intervention–this was a common occurrence for women not owning their own phone.

*R: Not able to listen to that [m-UPAVAN audios]. Husband takes and goes away. If want to listen to that thing a little, husband takes and goes away. So, at that time I get problems while listening to that.*

Other emerging themes showed that women were motivated to watch or listen due to curiosity about what future information videos or audios may entail, encouragement by facilitators, and eagerness to participate in the quizzes that followed each weekly video or audio dissemination. In contrast, some women expressed being nervous or fearful of participating in quizzes, either due to a lack of confidence using the phone or concern over what others may think if they were to respond.

**Table 3. Adjusted differences in feasibility and other outcomes across women's and household characteristics.**

| Indicator | N | High *vs* low education Mean or pp diff. (95% CI) | *p*-value | Non-ST *vs* ST Mean or pp diff. (95% CI) | *p*-value | Women owning their own phone *vs* not Mean or pp diff. (95% CI) | *p*-value | Smartphone *vs* no smartphone in household Mean or pp diff. (95% CI) | *p*-value |
|---|---|---|---|---|---|---|---|---|---|
| *Feasibility* | | | | | | | | | |
| Received videos or audios at least once, pp diff | 981 | 2.5 (-4.1, 9.1) | 0.46 | 0.6 (-6.5, 5.4) | 0.85 | 9.5 (2.7, 16.3) | 0.006 | 9.4 (3.2, 15.6) | 0.003 |
| Watched or listened at least once, pp diff | 981 | 1.4 (-5.3, 8.3) | 0.68 | 3.3 (-2.8, 9.5) | 0.29 | 7.2 (0.28, 14.1) | 0.04 | 1.6 (-0.5, 0.8) | 0.62 |
| *Other outcomes* | | | | | | | | | |
| Total videos or audios watched or listened to, mean diff | 981 | 0.10 (-0.09, 0.29) | 0.31 | 0.06 (-0.10, 0.23) | 0.44 | 0.25 (0.07, 0.43) | 0.007 | 0.13 (-0.04, 0.29) | 0.14 |
| Spouse or senior family member also watched or listened [a], pp diff | 710 | -8.5 (-17.3, 0.3) | 0.06 | -1.2 (-9.1, 6.7) | 0.77 | -5.8 (-14.5, 3.0) | 0.20 | 9.9 (1.8, 18.0) | 0.02 |
| Discussed video or audio content with adult family member [a], pp diff | 710 | 4.0 (-5.1, 13.0) | 0.39 | 1.7 (-6.2, 9.7) | 0.67 | 5.1 (-3.8, 14.0) | 0.26 | 2.4 (-5.7, 10.5) | 0.57 |

| Predicted % or mean in each sub-group | | Education | | Social group | | Mother owns a phone | | Household owns a smartphone | |
|---|---|---|---|---|---|---|---|---|---|
| | N | High | Low | Non-ST | ST | Yes | No | Yes | No |
| Received videos or audios at least once, % | 981 | 73.8 | 71.3 | 72.7 | 73.3 | 75.8 | 66.3 | 76.3 | 66.9 |
| Watched or listened at least once, % | 981 | 71.5 | 70.1 | 72.7 | 69.3 | 73.1 | 65.9 | 71.6 | 70.0 |
| Total videos or audios watched or listen to, mean | 981 | 1.6 | 1.5 | 1.7 | 1.6 | 1.7 | 1.4 | 1.7 | 1.5 |
| Spouse or senior family member also watched or listened [a], % | 710 | 58.4 | 66.9 | 60.2 | 61.4 | 59.3 | 65.1 | 64.1 | 54.1 |
| Discussed video or audio content with adult family member [a], % | 710 | 52.7 | 48.7 | 52.3 | 50.6 | 52.8 | 47.8 | 52.3 | 49.9 |

Note: PP = Percentage point; ST = Scheduled Tribe group; non-ST = Other caste groups. The recall period for all outcomes for each survey round is the previous 4 weeks. High *vs* low education compares women who completed primary school (≥8 years of schooling) to those who did not. Percentage and mean differences are average adjusted marginal predictions obtained from mixed-effects logistic or linear regression. Models are adjusted for all other covariates in the table and include random effects to account for repeated measures within individuals (levels 1) and within clusters (level 2). [a] Only includes participants reporting watching or listening at least once in the previous 4 weeks.

*I: What motivated you to reply [to quizzes]? R: As I found the videos helpful I gave a reply. Madam [facilitator] also encouraged me to give a reply [. . .] when I know the answer to a question then I get excited to reply [. . .].*

*R: I wanted to press those buttons 1 or 2 or 3 but was unable to do that I: Why were you unable to do that? R: Because I do not know how to do that. I: [. . .] Why did you not ask anyone if you were not able to understand? [. . .] R: I have asked once. She [facilitator] told me to press the button and to talk with them. . .I have not done that. I: Is there any reason why you have not? R: [. . .] Because of some fear I have not tried.*

## Acceptability

All respondents who watched or listened reported liking the m-UPAVAN messages. A favoured aspect was the mobile delivery platform, which enabled women to gain information from the comfort of their homes and at a convenient time. Women highlighted how the ability to re-watch the videos to reinforce their learning was also important to them. However, this benefit was less applicable to the audios, as audios were only available for a week.

*R: [. . .] Thanks to mobile and WhatsApp, I can take part in all those, right? If it needed us to go somewhere then we couldn't have gone that much. As the mobile is there, everything comes on WhatsApp, there's more interest, we can watch it all and learn it all.*

The content of the m-UPAVAN messages was praised highly by respondents, who often mentioned their favourite topics, which varied from nutrition-specific videos to ivy gourd farming, with the video on making pot manure fertiliser most frequently highlighted. The content revealed popularity due to the associated monetary benefits, as the promoted agriculture practices aimed to improve productivity and diversity. Frequently, though, monetary benefits were framed as desirable because they contributed towards improving the nutrition and health of women and their families. Several respondents also highlighted how the intervention content helped address taboos around women and children consuming certain foods.

*I: What are the effects of this* [m-UPAVAN] *on you, your family and your kids? [. . .] R: [. . .] We are farming ourselves and then giving them [crops] to our kids and are eating ourselves. We are staying healthy. We are farming small amounts and selling them to buy essentials. We are eating all that. And both mother and kids are remaining healthy.*

*R: I got to know that we [pregnant women] can eat anything we want. The elders used to say that we shouldn't eat more during pregnancy. But now we have got to know that we can eat more, and we should not do heavy work during pregnancy. We should take rest.*

Additionally, respondents felt the knowledge they gained was unique, and otherwise inaccessible if they did not watch the videos or listen to the audio messages. Respondents felt empowered with this knowledge.

*I: What was your first reaction after watching videos for the first time?*

*R: I became happy after watching. We generally have no idea regarding this, but due to this program we got to know about many things like plantation, health, nutrition, proper food for pregnant woman.*

## Intensity of uptake

Whilst 71% of respondents watched or listened to at least one video or audio in the past month, women only watched or listened to a mean of 1.61 (95% CI 1.50, 1.72), out of a maximum of 4 messages per month and women listened to fewer audios (0.52, 95% CI 0.45, 0.59) than videos (1.36, 95% CI 1.25, 1.47) (**Table 2**). Intensity of uptake overall (videos and audios) only varied between women owning a phone *vs* not (adjusted mean difference (95% CI): 0.25, 0.07, 0.43) (**Table 3**). Qualitative findings explained how, whilst women without their own phones could engage with the intervention occasionally, it was too infrequent, not at their convenience, and they were not able to re-watch the content as they wished.

*R: I watch as per my free time, but I cannot watch as much as I want. I: Why? R: Because it is their mobile, right? Will they always give me to watch on their mobile? I watch it once, but even if I am interested to watch it a second time I do not get to watch it on mobile.*

## Engaging family members

The mobile platform of the m-UPAVAN intervention offered an opportunity to engage whole families, and facilitators played a critical role in encouraging this. One woman explained the encouragement she received from her local facilitator:

*R: He'd [facilitator] say: "[. . .] show those videos to your family members and observe whether they are getting interested or not and you should also make them understand what is explained in the video because they will think "watching these videos, what will I get?" but you have to make them understand that it will be beneficial, tell them what benefits you have got by watching these videos so that they will also watch the videos" like this he used to encourage us.*

Quantitative findings showed the facilitators' encouragement was successful. Of those who watched or listened, 60.8% (95% CI 56.3, 65.3) reported their spouses or other senior household members also watching or listening. Whilst families engaging with content was higher among women from smartphone-owning households (adjusted pp difference: 9.9, 1.8, 18.0) (**Table 3**), over 50% of women from households without a smartphone still reported their spouses or senior family members listening. Women expressed how it was important for their families to learn directly from the videos and audios to ensure they received accurate information and could subsequently support women in implementing new behaviours.

*R: They [respondent's family] like the videos. If I had seen the videos alone, I might not have been able to share everything. They wouldn't have been able to know everything.*

*I: Do you think that they should participate in the m-UPAVAN program? R: Yes. I: Why so? R: Because they will understand it better and will help me in doing those things. Their health will also be good.*

Around half of the women watching or listening also reported discussing the content with other family members (**Table 2**), and whether these discussions took place did not vary across different characteristics and phone ownership (**Table 3**). The respondents explained that family members learning from m-UPAVAN messages directly helped ease family-level discussions and negotiations around the agriculture and nutrition practices promoted and facilitated cooperation among family members.

*I: So do you find explaining it [information from the videos] to your family members to be easy or difficult?*

*R: It is easy as they themselves see the video.*

*R: I learnt how to send a message on the phone and how to discuss things with family members. We got to know all these through listening to the talks in m-UPAVAN.*

*R: All the videos were doable. We don't need to invest a lot of money to implement those videos. It just needs cooperation of family members.*

### Preference for in-person UPAVAN or mobile-based m-UPAVAN

Overall, most participants highlighted various aspects that they liked about both the mobile and face-to-face intervention, with only a few participants providing a definite answer of their 'favourite.' Aspects of m-UPAVAN that were preferred related to being able to learn from the comfort of their own homes, with little opportunity cost concerning time trade-offs, and being able to refer back to messages for 'revision'. Simultaneously, participants found the longer video format in UPAVAN more conducive to learning and easier to understand, especially for those with less education and digital literacy.

> R: *In [m-UPAVAN] we can listen to it at our convenience and after 2–3 days if we forget any-thing we can listen to it again if we want. But in UPAVAN, [facilitator] would come and show us and then the next time there will be a new video.*

> R: *In UPAVAN among 20 mothers, we watch [and discuss] videos for around 1 to 1.5 hours. The videos in m-UPAVAN are for short duration so I don't like it much. If they would start UPAVAN again I will like it.*

Additionally, the mobile intervention did not adequately replace other aspects of the UPA-VAN face-to-face group-based approach. Many women highlighted the importance of discuss-ing in a group, sharing opinions with other women, and the enjoyment they felt from social interactions. Women who had taken part in the participatory learning and action meetings of UPAVAN often referred fondly to the elements designed for collective problem-solving and action. Further, equity concerns about m-UPAVAN were commonly expressed, as the oppor-tunity to learn was unequal between women with and without phone access and with smart-phones versus feature phones.

> R: *In UPAVAN we get to watch the videos, and after that, they conduct the meeting as well and everyone gets to share their opinions. That's why I like UPAVAN, even they make us play games* [. . .] *Now there is no one to discuss with and everything remains within our family members,* if [UPAVAN] *is conducted again, all of us mothers would be happy [. . .] For m-UPAVAN not everyone has a phone, they can only get to know through others.*

> R: *In UPAVAN we are shown the videos in a very easy way but in m-UPAVAN to see the video one has to have a smartphone and data pack all the time. But not everyone can afford that. So UPAVAN is a lot better and more effective in that way.*

Despite this, the same woman quoted above who liked UPAVAN more acknowledged that m-UPAVAN motivated her more to implement the methods, and was important for enabling her family to learn too. Yet, not all women felt this way. Some women expressed how UPA-VAN was necessary for understanding 'properly' and then subsequently being able to imple-ment what was learned.

> I: *Which one did motivate you to implement the methods? UPAVAN or m-UPAVAN? R: m-UPAVAN [because] in m-UPAVAN we can watch the video and implement it simulta-neously. We can watch the video again and again. Once, I was watching a video on diarrhoea of children. My older son saw that and asked me about it and I explained to him. He is very curious to know different things. So are my family members. So, in m-UPAVAN, not only do I get to know, but the whole family learns. My husband also supports this program as it gives a lot of information.*

> I: *[. . .] Which program will you recommend to people to participate in [m-UPAVAN or UPAVAN]?*

> R: *I will recommend participating in UPAVAN.*

> I: *Why?*

> R: *In UPAVAN they will see directly, they will understand it and they can apply it after under-standing it properly, so I will suggest they watch UPAVAN.*

Collectively, women's reflections pointed towards the necessity of both intervention modalities.

*R: I like both [m-UPAVAN and UPAVAN]. Both are important and enjoyable in their own dimensions.*

*R: I wish UPAVAN could be started again but still m-UPAVAN is helpful as we can watch the video again and again in case we forget any procedure. So both are interesting.*

## Discussion

We aimed to understand whether delivering NSA and nutrition-specific social behaviour change communication in the form of participatory WhatsApp videos and IVR audios offers a potential approach for improving agriculture and maternal and child nutrition in rural Odisha, India. During and since COVID-19, there has been an increased need and interest in remote delivery of agriculture, nutrition and health services [21,22]. However, to our knowledge, this is the first study to examine how a mobile adaptation of an NSA intervention that improved maternal and child diets could work in rural, disadvantaged, agrarian communities.

Among the mothers that had intra-household phone access, m-UPAVAN was demonstrated as a feasible and acceptable NSA intervention with our pre-specified targets for coverage, uptake, and acceptability met. We found that the interpersonal communication provided by the well-trained and trusted local intervention facilitators was integral for community mobilisation and engaging women and their families with the intervention. A wider body of research demonstrates that mobilising communities and delivering quality services requires eliciting and sustaining trust between those delivering the health services, e.g., community health workers and the community [23,24]. Our study showed how this continues to remain of critical importance even when using digital approaches. This finding is consistent with that from a digital messaging intervention in Telangana, India, delivered during COVID-19, which showed how digital approaches offer opportunity to improve the reach and consistency of behaviour change communication but complement rather than replace in-person approaches [25].

Throughout COVID-19, as mobile phones became the pivotal platform for information sharing and delivering health interventions, the need for flexible intervention delivery platforms became increasingly important. This and other studies taught us that, whilst mobile approaches are feasible, equity issues cannot be overlooked [5]. Social norms regarding women accessing and using phones are concerns for mHealth services in low-middle-income countries [26]. Whilst we found that women generally displayed agency using phones, digital gender inequality was greater among households that owned just one phone as men's phone usage was prioritised, restricting women's access to the intervention. Although we only found feasibility and acceptability outcomes to vary by phone ownership type, it is well known that the most marginalised groups and least educated, those with the greatest needs for nutrition and health services, have worse phone access [27]. Further, the most vulnerable women were likely among the 30% of women excluded from our quantitative feasibility and acceptability assessment due to no phone access. Additionally, whilst our objective was not to compare feasibility and acceptability between m-UPAVAN videos and audios, the lower engagement with the audio messages than videos implies the audio messages were a less feasible intervention strategy. This may have been due to the audios being less engaging than the videos and the more active participation required (women had to call to receive audios, whilst videos were automatically sent to them). As such, women who only have access to a feature phone, likely

poorer and more nutritionally vulnerable groups [27], may benefit less from the intervention than those with access to a smartphone.

Despite these findings, our study revealed how mobile interventions can overcome barriers related to face-to-face approaches. A favourite aspect of m-UPAVAN was the time women saved from travelling and the ability to learn on-demand. A family-centric virtual counselling mHealth intervention tested in Nepal demonstrates the importance of the on-demand nature of m-UPAVAN [28]. In addition to other barriers related to poor network connectivity and digital literacy, the evaluation found that scheduling the counselling services was challenging for some women due to their competing time demands, and family members were hard to engage due to working outside of the home [28]. Whether women are able to participate in NSA programmes and thus benefit from them may be influenced by time trade-offs [29], and the time to participate in programmes is not equally available to all women [30]. As such, the time poverty of women must not be overlooked when designing interventions for marginalised communities, especially as poorer women tend to be more affected [31]. On-demand mobile interventions may offer opportunities to alleviate some of these concerns, albeit unlikely for the most marginalised groups. Additionally, although we found that the longer videos and facilitated discussions of the UPAVAN interventions were more conducive to learning, women expressed how the ability to refer back to videos to revise the content minimised information loss, which may enhance the likelihood of implementing promoted practices. Taken together, these findings suggest a need for a hybrid intervention that offers interpersonal interaction and participatory elements to promote peer-to-peer learning and sharing approaches, whilst also offering flexibility, being considerate of women's limited time, and giving opportunities to revise and retain knowledge through the mHealth approach.

A further benefit of the mobile platform was the successful engagement of whole families. Lack of family support is a key barrier to accessing nutrition interventions and adopting promoted behaviours in low and middle-income countries [32] and family support and cooperation were identified as important enabling factors in the UPAVAN trial [10,33]. Following this, m-UPAVAN actively promoted watching and listening with family members via phone messages and with facilitators after COVID-19 restrictions eased. Over 60% of women engaging with m-UPAVAN content reported their spouse or another senior family member also doing so, and around half discussed the content with their families. Engaging whole families enabled discussions and negotiations around agriculture and nutrition practices promoted in the videos, which mothers believed facilitated their learning and adoption of behaviours. More active and deliberate engagement with other family members could help further increase the effectiveness of interventions [34,35]. For example, mobile interventions in India have proved successful in improving men's knowledge of maternal health through text messaging [36]. Further research is needed to understand whether approaches that include other household members yield greater agriculture and nutrition impacts than those predominantly focused on women and the potential mediating mechanisms. However, strategies that engage whole families inclusively, i.e., not excluding those without phone access, also need developing and testing.

Whilst making interventions more family-centric may be key to enhancing nutritional impacts, women's group-based learning remains important and was often highlighted as why mothers preferred UPAVAN over m-UPAVAN. Group-based approaches focused on social interactions create spaces for women to share experiences and gain confidence,where new behaviours are learnt and collectively adopted [37]. In India, women's self-help groups and participatory learning and action groups (including those used in UPAVAN), to which learning and action-based interpersonal social interactions are central and empowering to communities, have proven effective at improving diets and other health outcomes [38–41].

The design of the m-UPAVAN intervention incorporated key learnings from the UPAVAN process evaluation, which found that lack of time, water, land, and family support meant that many families only adopted a limited number of practices, despite the intervention addressing these [10]. As such, to ensure participants could gauge the relevance of the upcoming messages, m-UPAVAN facilitators actively promoted the audio and videos before they were disseminated including providing information on the various constraints that the messages addressed (e.g., lack of time, water or land). To alleviate barriers arising from a lack of family support, facilitators actively encouraged women to watch or listen with their family members. Resulting, we found that m-UPAVAN was highly accepted, with participants expressing that the promoted practices were useful and feasible based on available resources. However, whilst this positive feedback suggests a need to evaluated the impact of the mobile intervention on agriculture and nutrition outcomes, it is important to note that our study sample likely represents a better-off portion of the population. m-UPAVAN was initiated during the COVID-19 lockdown in Odisha when only phone surveys were feasible. As such, over a quarter of women belonging to households without phones were excluded from our quantitative surveys. Therefore, our study sample may generally face fewer constraints in adopting promoted practices relative to the wider population.

Our study's findings underpin the necessity of addressing inequities in phone access and use to optimise mHealth interventions' potential in low- and middle-income settings. Training and capacity building on phone and mobile app usage in rural communities could help increase the uptake of mHealth interventions and alleviate gender-based social norms in phone usage [42]. Many policy initiatives are being implemented across low and middle-income settings [43,44], including in India [45,46], to improve access to digital technologies in rural and disadvantaged communities. Future mHealth interventions and policy approaches should work closely with relevant initiatives to increase their uptake and effectiveness. mHealth interventions that incorporate a microcredit component to help certain individuals purchase phones may also be a feasible approach to reducing inequities in mHealth [42]. Future research must explicitly evaluate the effects of digital inequality on the efficacy and equity of future mHealth interventions.

## Limitations and strengths

A limitation of our study is sampling bias and non-response bias. As women without a phone were excluded from quantitative surveys and monitoring survey respondents were economically better off than non-respondents, the reported feasibility and acceptability of the intervention may have been inflated relative to a sample representative of all women with a child under 2 years in study villages. Whilst unfeasible at the time of our study due to COVID-19 restrictions, evaluations of future digital interventions should aim to use the most inclusive methods available for recruiting and interviewing participants to overcome these biases, such as in-person interviews or using on-the-ground resource persons to help facilitate the participation of harder-to-reach groups in mobile surveys. Additionally, social desirability may have resulted in respondents overreporting positive attitudes towards the intervention. However, quantitative surveys administered by an independent data collection firm, which was not involved in the intervention, aimed to minimise this risk. A key strength is our mixed-methods approach, which enabled us to understand intervention feasibility and acceptability and unpack the factors that explained it. Second, delivering the adapted intervention in the same study area and within a short time frame of the original intervention provided a unique opportunity to understand preferences regarding the in-person *vs.* mobile-based intervention.

## Conclusion

Mobile phones can provide a feasible and acceptable way to deliver NSA interventions among those with phone, especially smartphone, access, and can overcome several barriers related to in-person approaches: provide NSA services on demand, reinforce learning, and engage whole families. However, despite mobile coverage growing rapidly in rural India [47], equity issues remain. Participatory face-to-face approaches continue to be essential for improving the equity of NSA interventions and are more conducive to learning. There is a need to develop and test the effectiveness of hybrid NSA interventions that incorporate the complementary strengths of both mobile and face-to-face interventions for improving agricultural and nutrition outcomes.

## Supporting information

**S1 Checklist. Inclusivity in global research.**
(PDF)

**S1 Table. Characteristics of purposely sampled qualitative interview participants (*N* = 38).** Participants in UPAVAN study areas who confirmed they participated in the UPAVAN interventions were administered open-ended questions regarding preferences for m-UPAVAN *vs* UPAVAN. All mothers, besides those belonging to a household without a phone, were recruited from those who participated in monthly monitoring surveys.
(DOCX)

**S2 Table. Nutrition-sensitive agriculture and nutrition-specific topics disseminated in m-UPAVAN.**
(DOCX)

**S1 Appendix. Village-wise poster that advertised the toll-free line for accessing m-UPAVAN audios.**
(DOCX)

**S2 Appendix. Semi-structured interview guides used in qualitative interviews.**
(DOCX)

## Acknowledgments

The authors would like to thank all the women and their families who participated in the interventions, responded to surveys, and took part in qualitative interviews. We also thank the Voluntary Association for Rural Reconstruction and Appropriate Technology field staff, who dedicated their time to delivering the interventions, and DCOR survey staff for collecting quantitative data and conducting qualitative interviews.

## Author Contributions

**Conceptualization:** Emily Fivian, Manoj Parida, Helen Harris-Fry, Satyanarayan Mohanty, Shibanath Padhan, Ronali Pradhan, Audrey Prost, Philip James, Suneetha Kadiyala.

**Data curation:** Emily Fivian.

**Formal analysis:** Emily Fivian, Pranay Das, Gladys Odhiambo.

**Funding acquisition:** Suneetha Kadiyala.

**Investigation:** Emily Fivian, Manoj Parida, Satyanarayan Mohanty, Shibanath Padhan, Ronali Pradhan, Satyaranjan Behera, Naba Kishor Mishra.

**Methodology:** Suchitra Rath, Nirmala Nair, Shibanand Rath, Peggy Koniz-Booher, Heather Danton.

**Project administration:** Emily Fivian, Shibanath Padhan, Ronali Pradhan, Naba Kishor Mishra, Suneetha Kadiyala.

**Software:** Satyaranjan Behera.

**Supervision:** Terry Roopnaraine, Elizabeth Allen, Suneetha Kadiyala.

**Writing – original draft:** Emily Fivian.

**Writing – review & editing:** Emily Fivian, Manoj Parida, Helen Harris-Fry, Gladys Odhiambo, Audrey Prost, Terry Roopnaraine, Philip James, Peggy Koniz-Booher, Elizabeth Allen, Suneetha Kadiyala.

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
