## [Decision Letter · Decision Letter 0]

10 Jan 2024

PGPH-D-23-01837

Feasibility, acceptability and equity of a mobile intervention for Upscaling Participatory Action and Videos for Agriculture and Nutrition (m-UPAVAN) in rural Odisha, India.

Dear Dr. Fivian,

Thank you for submitting your manuscript to PLOS Global Public Health. After careful consideration, we feel that it has merit but does not fully meet PLOS Global Public Health’s publication criteria as it currently stands. Therefore, we invite you to submit a revised version of the manuscript that addresses the points raised during the review process.

Please note that we have only been able to secure a single reviewer to assess your manuscript. We are issuing a decision on your manuscript at this point to prevent further delays in the evaluation of your manuscript. Please be aware that the editor who handles your revised manuscript might find it necessary to invite additional reviewers to assess this work once the revised manuscript is submitted. However, we will aim to proceed on the basis of this single review if possible. 

The manuscript has been evaluated by one reviewer, and their comments are available below.

The reviewer has raised a number of concerns that need attention. Specifically, they are requesting you provide a more detailed destinction between the initial UPAVAN and your proposed m-UPAVAN intervention.

Could you please revise the manuscript to carefully address the concerns raised?

We look forward to receiving your revised manuscript.

Kind regards,

Johanna Pruller, Ph.D.

PLOS Staff Editor

Journal Requirements:

Additional Editor Comments (if provided):

Reviewers' comments:

Reviewer's Responses to Questions

**Comments to the Author**

1. Does this manuscript meet PLOS Global Public Health’s publication criteria? Is the manuscript technically sound, and do the data support the conclusions? The manuscript must describe methodologically and ethically rigorous research with conclusions that are appropriately drawn based on the data presented.

Reviewer #1: Yes

2. Has the statistical analysis been performed appropriately and rigorously?

Reviewer #1: Yes

3. Have the authors made all data underlying the findings in their manuscript fully available (please refer to the Data Availability Statement at the start of the manuscript PDF file)?

Reviewer #1: Yes

4. Is the manuscript presented in an intelligible fashion and written in standard English?

Reviewer #1: Yes

5. Review Comments to the Author

Reviewer #1: The manuscript is a well conceived idea that would ultimately improve the well-being of the respondents in rural contexts. The comments below are given to improve the study:

Abstract: The authors do not provide the results in a readable and logical way. As a reader it will be great to see the abstract (especially results) written in a phased manner.

Line 163: the role and contribution of Digital Green is somewhat unclear. Did they participate in developing the model for initial UPAVAN. If no, then what qualifies them to be a creator for m-UPAVAN in this case.

The authors previously asserted the composition of UPAVAN's content (line 75-83) with its goals. But this not clear with m-UPAVAN. It seems that m-UPAVAN applies the same principles like UPAVAN but not same content. This distinction has to be clearly stated. The decision to also use the word m-UPAVAN thus becomes questionable.

The selected rates decided by the study (line 221-240) seem ambiguous. It is expected that such cut-offs will be selected based on literature or a more scientific approach will be used to ascertain the cut-offs.

6. PLOS authors have the option to publish the peer review history of their article (what does this mean?). If published, this will include your full peer review and any attached files.

**Do you want your identity to be public for this peer review?** For information about this choice, including consent withdrawal, please see our Privacy Policy.

Reviewer #1: No

---

## [Decision Letter · Decision Letter 1]

4 Apr 2024

PGPH-D-23-01837R1

Feasibility, acceptability and equity of a mobile intervention for Upscaling Participatory Action and Videos for Agriculture and Nutrition (m-UPAVAN) in rural Odisha, India.

Dear Dr. Emily%,

Thank you for submitting your manuscript to PLOS Global Public Health. After careful consideration, we feel that it has merit but require minor revisions. Therefore, we invite you to submit a revised version of the manuscript that addresses the points raised during the review process.

We look forward to receiving your revised manuscript.

Kind regards,

Jyoti Sharma

Academic Editor

Journal Requirements:

Additional Editor Comments (if provided):

Thanks for your submission. the second reviewer reviewed your manuscript and suggested Minor revision. Pl. the detailed comments.

requesting you to pl. revise the manuscript in view of these comments.

Reviewers comments

1. Thank you for coming up with an excellent intervention. I would like to see a bit more analysis, especially the authors’ perspectives on how to overcome digital poverty among rural women of LMICs. Any future direction or research ideas other than mixed kind intervention (face to face and digital) will be a great addition in the manuscript.

2. You mentioned about bias of socio-economic hierarchy in selecting participants. Can you provide suggestions regarding overcoming this biases in future digital interventions?

3. Methodology section is a great read. Please add your reflexivity statements as researchers working on rural context and during telephonic interactions with the participants.

Reviewers' comments:

Reviewer's Responses to Questions

**Comments to the Author**

1. If the authors have adequately addressed your comments raised in a previous round of review and you feel that this manuscript is now acceptable for publication, you may indicate that here to bypass the “Comments to the Author” section, enter your conflict of interest statement in the “Confidential to Editor” section, and submit your "Accept" recommendation.

Reviewer #1: All comments have been addressed

Reviewer #2: All comments have been addressed

2. Does this manuscript meet PLOS Global Public Health’s publication criteria? Is the manuscript technically sound, and do the data support the conclusions? The manuscript must describe methodologically and ethically rigorous research with conclusions that are appropriately drawn based on the data presented.

Reviewer #1: Yes

Reviewer #2: Yes

3. Has the statistical analysis been performed appropriately and rigorously?

Reviewer #1: N/A

Reviewer #2: Yes

4. Have the authors made all data underlying the findings in their manuscript fully available (please refer to the Data Availability Statement at the start of the manuscript PDF file)?

Reviewer #1: Yes

Reviewer #2: Yes

5. Is the manuscript presented in an intelligible fashion and written in standard English?

Reviewer #1: Yes

Reviewer #2: Yes

6. Review Comments to the Author

Reviewer #1: Nil

Reviewer #2: 1.Thank you for coming up with an excellent intervention. I would like to see a bit more analysis, especially the authors’ perspectives on how to overcome digital poverty among rural women of LMICs. Any future direction or research ideas other than mixed kind intervention (face to face and digital) will be a great addition in the manuscript.

2.You mentioned about bias of socio-economic hierarchy in selecting participants. Can you provide suggestions regarding overcoming this biases in future digital interventions?

3.Methodology section is a great read. Please add your reflexivity statements as researchers working on rural context and during telephonic interactions with the participants.

7. PLOS authors have the option to publish the peer review history of their article (what does this mean?). If published, this will include your full peer review and any attached files.

**Do you want your identity to be public for this peer review?** For information about this choice, including consent withdrawal, please see our Privacy Policy.

Reviewer #1: No

Reviewer #2: **Yes: **Dr. Upasona Ghosh

---

## [Editor Report · Decision Letter 2]

17 Apr 2024

Feasibility, acceptability and equity of a mobile intervention for Upscaling Participatory Action and Videos for Agriculture and Nutrition (m-UPAVAN) in rural Odisha, India.

PGPH-D-23-01837R2

Dear Ms Fivian,

We are pleased to inform you that your manuscript 'Feasibility, acceptability and equity of a mobile intervention for Upscaling Participatory Action and Videos for Agriculture and Nutrition (m-UPAVAN) in rural Odisha, India.' has been provisionally accepted for publication in PLOS Global Public Health.

Best regards,

Jyoti Sharma

Academic Editor